# Semi-Parametric Dynamic Contextual Pricing

**Virag Shah**
Management Science and Engineering
Stanford University
California, USA 94305
virag@stanford.edu

**Jose Blanchet**
Management Science and Engineering
Stanford University
California, USA 94305
jblanche@stanford.edu

**Ramesh Johari**
Management Science and Engineering
Stanford University
California, USA 94305
rjohari@stanford.edu

## Abstract

Motivated by the application of real-time pricing in e-commerce platforms, we consider the problem of revenue-maximization in a setting where the seller can leverage contextual information describing the customer's history and the product's type to predict her valuation of the product. However, her true valuation is unobservable to the seller, only binary outcome in the form of success-failure of a transaction is observed. Unlike in usual contextual bandit settings, the optimal price/arm given a covariate in our setting is sensitive to the detailed characteristics of the residual uncertainty distribution. We develop a semi-parametric model in which the residual distribution is non-parametric and provide the first algorithm which learns both regression parameters and residual distribution with $\tilde{O}(\sqrt{n})$ regret. We empirically test a scalable implementation of our algorithm and observe good performance.

## 1 Introduction

Many e-commerce platforms are experimenting with approaches to personalized dynamic pricing based on the customer's *context* (i.e. customer's prior search/purchase history and the product's type). However, the mapping from context to optimal price needs to be learned. Our paper develops a bandit learning approach towards solving this problem motivated by practical considerations faced by online platforms. In our model, customers arrive sequentially, and each customer is interested in buying one product. The customer purchases the product if her *valuation* (unobserved by the platform) for the product exceeds the price set by the seller. The platform observes the covariate vector corresponding to the context, and chooses a price. The customer buys the item if and only if the price is lower than her valuation.

We emphasize three salient features of this model; taken together, these are the features that distinguish our work. *First*, feedback is only binary: either the customer buys the item, or she does not. In other words, the platform must learn from censored feedback. This type of binary feedback is a common feature of practical demand estimation problems, since typically exact observation of the valuation of a customer is not possible.

*Second*, the platform must learn the functional form of the relationship between the covariates and the expected valuation. In our work, we assume a parametric model for this relationship. In particular, we presume that the expected value of the logarithm of the valuation is linear in the covariates. Among

other things, this formulation has the benefit that it ensures valuations are always nonnegative. Further, from a technical standpoint, we demonstrate that this formulation also admits efficient estimation of the parametric model.

*Third*, the platform must also learn the distribution of residual uncertainty that determines the actual valuation given the covariates; in other words, the distribution of the *error* between the expected logarithm of the valuation, and the actual logarithm of the valuation, given covariates. In our work we make minimal assumptions about the distribution of this residual uncertainty. Thus while the functional relationship between covariates and the expected logarithm of the valuation is parametric (i.e., linear), the distribution of the error is nonparametric; for this reason, we refer to our model as a *semi-parametric* dynamic pricing model.

The challenge is to ensure that we can efficiently learn both the coefficients in the parametric model, as well as the distribution of the error. A key observation we leverage is that our model exhibits *free exploration*: testing a single covariate-vector-to-price mapping at a given time can simultaneously provide information about *several* such mappings. We develop an arm elimination approach which maintains a set of active prices at each time, where the set depends on the covariate vector of the current customer. The set is reduced over time by eliminating empirically suboptimal choices.

We analyze our approach both theoretically and empirically. We analyze regret against the following standard oracle: the policy that optimally chooses prices given the true coefficients in the parametric linear model, as well as the distribution of the error, but without knowledge of the exact valuation of each arriving customer. Regret of our policy scales as $\tilde{O}(\sqrt{n})$ with respect to time horizon $n$, which is optimal. Further, it scales polynomially in covariate dimension $d$, as well as in two smoothness parameters $\kappa_1$ and $\kappa_2$ defined as part of our model. In addition, we develop a scalable implementation of our approach which leverages a semi-parametric regression technique based on convex optimization. Our simulations show that this scalable policy performs well.

## 1.1 Related work

**Non-contextual dynamic pricing.** There is a significant literature on regret analysis of the dynamic pricing problem without covariates; see den Boer (2015) for a detailed survey. For example, the works Le Guen (2008); Broder and Rusmevichientong (2012); den Boer and Zwart (2013); den Boer (2014); Keskin and Zeevi (2014) consider a parametric model whereas Kleinberg and Leighton (2003) consider a non-parametric model for the unknown demand function. Our methodology is most aligned to that of Kleinberg and Leighton (2003), in that we extend their techniques to incorporate side-information from the covariates.

**Contextual dynamic pricing.** Recently, the problem of dynamic pricing with high-dimensional covariates has garnered significant interest among researchers; see, e.g., Javanmard and Nazerzadeh (2019); Ban and Keskin (2019); Cohen et al. (2016b); Mao et al. (2018); Qiang and Bayati (2019); Nambiar et al. (2019). In summary, in contrast to the prior works in dynamic pricing with covariates, ours is the first work to address a setting where the only feedback from each transaction is binary and the residual uncertainty given covariates is non-parametric, see Table 1. We believe that these features are relevant to several online platforms implementing dynamic pricing with high-dimensional covariates, and thus our work bridges a gap between the state-of-the-art in the academic literature and practical considerations.

**Learning techniques:** There is extensive prior work on high-dimensional contextual bandits, e.g., Langford and Zhang (2008); Slivkins (2011); Perchet and Rigollet (2013); Greenewald et al. (2017); Krishnamurthy et al. (2018); however, their techniques do not directly apply to our setup (in part due to the censored nature of feedback). Our work is also loosely related to the works on learning and auctions, e.g. Amin et al. (2014); Morgenstern and Roughgarden (2016). We leverage semi-parametric regression technique with binary feedback from Plan and Vershynin (2013) to reduce computational complexity of our algorithm.

There are some similarities between our work and the literature on bandits with side information, e.g., Mannor and Shamir (2011); Alon et al. (2013); Caron et al. (2012); Cohen et al. (2016a); Lykouris et al. (2018). For example, in their work too there is free exploration where testing for one arm reveals the reward information for a subset of arms, where the subset may be a function of the chosen action. However, there are some crucial differences. In particular, these works assume (a) a discrete set of arms, (b) the existence of a sequence of graphs indexed by time (possibly fixed) with the arms

|  | Contextual | Non-parametric residuals | Binary feedback |
|---|:---:|:---:|:---:|
| Kleinberg and Leighton (2003) |  | ✓ | ✓ |
| Javanmard and Nazerzadeh (2019) | ✓ |  | ✓ |
| Qiang and Bayati (2019) | ✓ | ✓ |  |
| Cohen et al. (2016b); Mao et al. (2018) | ✓ |  | ✓ |
| Ban and Keskin (2019) | ✓ | ✓ |  |
|  | ✓ |  | ✓ |
| Nambiar et al. (2019) | ✓ | ✓ |  |
| Our work | ✓ | ✓ | ✓ |

Table 1: This table compares our results with prior work along three dimensions: (1) incorporating contextual information; (2) modeling the distribution of residual uncertainty (given the context, where appropriate) as non-parametric; and (3) receiving only binary success/failure feedback from each transaction.

as its nodes, (c) the action involves pulling an arm, and at each time the reward at each neighbor of the pulled arm is revealed. However, in our setting, it is important to model the set of prices, and thus the set of covariate-vector-to-price mappings as described above, as a continuous set since a constant error in price leads to linear regret. While in our DEEP-C policy we discretize the set of covariate-vector-to-price mappings into a finite set of arms (which scale with time horizon), the above assumptions are still not met due to the following. Each arm in our setting corresponds to a subset of prices/actions. The subset of arms for which the reward is revealed at time $t$ depends on the covariate $x_t$, and the exact price $p_t$ from the above subset. Thus, the assumption of a pre-defined graph structure is not satisfied.

## 2 Preliminaries

In this section we first describe our model and then our objective, which is to minimize regret relative to a natural oracle policy.

### 2.1 Model

At each time $t \in \{1, 2, \ldots, n\}$, we have a new user arrival with covariate vector $X_t$ taking values in $\mathbb{R}^d$ for $d \geq 1$. Throughout the paper all vectors are encoded as column vectors. The platform observes $X_t$ upon the arrival of the user. The user's reservation value $V_t \in \mathbb{R}$ is modeled as

$$\ln V_t = \theta_0^\mathsf{T} X_t + Z_t', \tag{1}$$

where $\theta_0 \in \mathbb{R}^d$ is a fixed unknown parameter vector, and $Z_t'$ for $t \in \{1, 2, \ldots, n\}$ captures the residual uncertainty in demand given covariates.

Similar to the linear model $V_t = \theta_0^\mathsf{T} X_t + Z_t'$, this model is quite flexible in that linearity is a restriction only on the parameters while the predictor variables themselves can be arbitrarily transformed. However, our formulation additionally has the feature that it ensures that $V_t > 0$ for each $t$, a key practical consideration. We conjecture that unlike our model, the linear model $V_t = \theta_0^\mathsf{T} X_t + Z_t'$ does not admit a learning algorithm with $\tilde{O}(\sqrt{n})$ regret. This is due to censored nature of feedback, the structure of revenue as a function of price, and our non-parametric assumption on the distribution of $Z_t'$ as described below. Also, exponential sensitivity of the valuation with respect to covariate magnitudes can be avoided by using a logarithmic transformation of the covariates themselves. More generally, one may augment our approach with a machine learning algorithm which learns an appropriate transformation to fit the data well. In this paper, however, we focus on valuation model as given by (1).

Equivalently to (1), we have

$$V_t = e^{\theta_0^\intercal X_t} Z_t,$$

where $Z_t = e^{Z_t'}$. Thus, $Z_t > 0$ for each $t$.

The platform sets price $p_t$, upon which the user buys the product if $V_t \geq p_t$. Without loss of generality, we will assume the setting where users buy the product; one can equivalently derive exactly the same results in a setting where users are sellers, and sell the product if $V_t \leq p_t$. The revenue/reward at time $t$ is $p_t Y_t$ where $Y_t = \mathbb{1}_{V_t \geq p_t}$. We assume that $p_t$ is $\sigma\left(X_1, \ldots, X_{t-1}, X_t, Y_1, \ldots, Y_{t-1}, U_1, \ldots, U_t\right)$ measurable, where $U_t$ for each $t \geq 1$ is an auxiliary $U[0, 1]$ random variable independent of the sources of randomness in the past. In other words, platform does not know the future but it can use randomized algorithms which may leverage past covariates, current covariate, and binary feedback from the past.

The goal of the platform is to design a pricing policy $\{p_t\}_{t \in \{1, \ldots, n\}}$ to maximize the total reward

$$\Gamma_n = \sum_{t=1}^{n} Y_t p_t.$$

In this paper we are interested in the performance characterization of optimal pricing policies as the time horizon $n$ grows large.

We make the following assumption on statistics of $X_t$ and $Z_t$.

**A1** *We assume that $\{X_t\}_t$ and $\{Z_t\}_t$ are i.i.d. and mutually independent. Their distributions are unknown to the platform. Their supports $\mathcal{X}$ and $\mathcal{Z}$ are compact and known. In particular, we assume that $\mathcal{X} \subset \left[-\frac{1}{2}, \frac{1}{2}\right]^d$ and $\mathcal{Z}$ is an interval in $[0, 1]$.*

A1 can be significantly relaxed, as we discuss in Appendix E (both in terms of the i.i.d. distribution of random variables, and the compactness of their supports).

**A2** *The unknown parameter vector $\theta_0$ lies within a known, connected, compact set $\Theta \subset \mathbb{R}^d$. In particular, $\Theta \subset [0, 1]^d$.*

It follows from A1 and A2 that we can compute reals $0 < \alpha_1 < \alpha_2$ such that for all $(z, x, \theta) \in \mathcal{Z} \times \mathcal{X} \times \Theta$ we have

$$\alpha_1 \leq z e^{\theta^\intercal x} \leq \alpha_2.$$

Thus, the valuation at each time is known to be in the set $[\alpha_1, \alpha_2]$, and in turn the platform may always choose price from this set. Note also that, since $\mathcal{Z} \subset [0, 1]$, for each $(x, \theta) \in \mathcal{X}$, we have that $\alpha_1 \leq e^{\theta^\intercal x} \leq \alpha_2$.

## 2.2 The oracle and regret

It is common in multiarmed bandit problems to measure the performance of an algorithm against a benchmark, or Oracle, which may have more information than the platform, and for which the optimal policy is easier to characterize. Likewise, we measure the performance of our algorithm against the following Oracle.

**Definition 1** *The* Oracle *knows the true value of $\theta_0$ and the distribution of $Z_t$.*

Now, let

$$F(z) = z\mathbb{P}(Z_1 \geq z).$$

The following proposition is easy to show, so the proof is omitted.

**Proposition 1** *The following pricing policy is optimal for the* Oracle*: At each time $t$ set price $p_t = z^* e^{\theta_0^\intercal X_t}$ where $z^* = \arg\sup_z F(z)$.*

Clearly, the total reward obtained by the Oracle with this policy, denoted as $\Gamma_n^*$, satisfies $\mathbb{E}[\Gamma_n^*] = nz^*\mathbb{E}[e^{\theta_0^\intercal X_1}]$.

**Our goal: Regret minimization.** Given a feasible policy, define the regret against the Oracle as $R_n$:

$$R_n = \Gamma_n^* - \Gamma_n.$$

Our goal in this paper is to design a pricing policy which minimizes $\mathbb{E}[R_n]$ asymptotically to leading order in $n$.

## 2.3 Smoothness Assumption

In addition to A1 and A2, we make a smoothness assumption described below.

Let

$$r(z, \theta) = z\mathbb{E}\left[ e^{\theta^{\mathsf{T}} X_1} \mathbf{1}\left\{ Z_1 e^{\theta_0^{\mathsf{T}} X_1} > z e^{\theta^{\mathsf{T}} X_1} \right\} \right],$$

which can be thought of as the expected revenue of a single transaction when the platform sets price $p = z e^{\theta^{\mathsf{T}} x}$ after observing a covariate $X = x$. We impose the following assumption on $r(z, \theta)$.

**A3** *Let $\theta^{(l)}$ be the $l^{th}$ component of $\theta$, i.e., $\theta = (\theta^{(l)} : 1 \le l \le d)$. We assume that there exist $\kappa_1, \kappa_2 > 0$ such that for each $z \in \mathcal{Z}$ and $\theta \in \Theta$ we have*

$$\kappa_1 \max\left\{ (z^* - z)^2, \max_{1 \le l \le d} (\theta_0^{(\ell)} - \theta^{(l)})^2 \right\} \le r(z^*, \theta_0) - r(z, \theta) \le \frac{\kappa_2}{d+1} \|(z^* - z, \theta_0 - \theta)\|^2$$

*where $\|(z, \theta)\|^2 = \left( z^2 + \sum_{l=1}^{d} (\theta^{(l)})^2 \right)$.*

Recall that $F(z) = z\mathbb{P}(Z_1 \ge z)$. It follows from A1 and conditioning on $X_1$ that

$$r(z, \theta) = \mathbb{E}\left[ e^{\theta_0^{\mathsf{T}} X_1} F\left( e^{-(\theta_0 - \theta)^{\mathsf{T}} X_1} z \right) \right].$$

We will use this representation throughout our development.

Note that A3 subsumes that $(z^*, \theta_0)$ is the unique optimizer of $r(z, \theta)$. This is true if $z^*$ is the unique maximizer of $F(z)$ and that $\theta_0$ is identifiable in the parameter space $\Theta$.

Below we will also provide sufficient conditions for A3 to hold. In particular, we develop sufficient conditions which are a natural analog of the assumptions made in Kleinberg and Leighton (2003).

## 2.4 Connection to assumptions in Kleinberg and Leighton (2003)

The 'stochastic valuations' model considered in Kleinberg and Leighton (2003) is equivalent to our model with no covariates, i.e., with $d = 0$. When $d = 0$ the revenue function $r(z, \theta)$ is equal to $F(z)$. In Kleinberg and Leighton (2003) it is assumed that $\{Z_t\}$ are i.i.d., and that $F(z)$ has bounded support. Clearly A1 and A2 are a natural analog to these assumptions. They also assume that $F(z)$ has unique optimizer, and is locally concave at the optimal value, i.e., $F''(z^*) < 0$. We show below that a natural analog of these conditions are sufficient for A3 to hold.

Suppose that $(z^*, \theta_0)$ is the unique optimizer of $r(z, \theta)$. Also suppose that A1 and A2 hold. Then A3 holds if $r(z, \theta)$ is strictly locally concave at $(z^*, \theta_0)$, i.e., if the Hessian of $r(z, \theta)$ at $(z^*, \theta_0)$ exists and is negative definite. To see why this is the case, note that strict local concavity at $(z^*, \theta_0)$ implies that there exists an $\epsilon > 0$ such that the assumption holds for each $(z, \theta) \in \mathcal{B}_\epsilon(z^*, \theta_0)$ where $\mathcal{B}_\epsilon(z^*, \theta_0)$ is the $d + 1$ dimensional ball with center $(z^*, \theta_0)$ and radius $\epsilon$. This, together with compactness of $\mathcal{X}$ and $\Theta$, implies A3.

It is somewhat surprising that to incorporate covariates in a setting where $F$ is non-parametric, only minor modifications are needed relative to the assumptions in Kleinberg and Leighton (2003). For completeness, in the Appendix we provide a class of examples for which it is easy to check that the Hessian is indeed negative definite and that all our assumptions are satisfied.

## 3 Pricing policies

Any successful algorithm must set prices to balance price *exploration* to learn $(\theta_0, z^*)$ with *exploitation* to maximize revenue. Because prices are adaptively controlled, the outputs $(Y_t : t = 1, 2, \ldots, n)$

will *not* be conditionally independent given the covariates $(X_t : t = 1, 2, \ldots, n)$, as is typically assumed in semi-parametric regression with binary outputs (e.g., see Plan and Vershynin (2013)). This issue is referred to as *price endogeneity* in the pricing literature.

We address this problem by first designing our own bandit-learning policy, Dynamic Experimentation and Elimination of Prices with Covariates (DEEP-C), which uses only a basic statistical learning technique which dynamically eliminates sub-optimal values of $(\theta, z)$ by employing confidence intervals. At first glance, such a learning approach seems to suffer from the curse of dimensionality, in terms of both sample complexity and computational complexity. As we will see, our DEEP-C algorithm yields low sample complexity by cleverly exploiting the structure of our semi-parameteric model. We then address computational complexity by presenting a variant of our policy which incorporates sparse semi-parametric regression techniques.

The rest of the section is organized as follows. We first present the DEEP-C policy. We then discuss three variants: (a) DEEP-C with Rounds, a slight variant of DEEP-C which is a bit more complex to implement but simpler to analyze theoretically, and thus enables us to obtain $\tilde{O}(\sqrt{n})$ regret bounds; (b) Decoupled DEEP-C, which decouples the estimation of $\theta_0$ and $z^*$ and thus allows us to leverage low-complexity sparse semi-parametric regression to estimate $\theta_0$ but with the cost of $O(n^{2/3})$ regret; and (c) Sparse DEEP-C, which combines DEEP-C and sparse semi-parametric regression to achieve low complexity without decoupling to achieve the best of both worlds. We provide a theoretical analysis of the first variant, and use simulation to study the others.

While we discuss below the key ideas behind these three variants, their formal definitions are provided in Appendix B to save space.

### 3.1 DEEP-C policy

We now describe DEEP-C. As noted in Proposition 1, the Oracle achieves optimal performance by choosing at each time a price $p_t = z^* e^{\theta_0^\intercal X_t}$, where $z^*$ is the maximizer of $F(z)$. We view the problem as a multi-armed bandit in the space $\mathcal{Z} \times \Theta$. Viewed this way, *before* the context at time $t$ arrives, the decision maker must choose a value $z \in \mathcal{Z}$ and a $\theta \in \Theta$. Once $X_t$ arrives, the price $p_t = z e^{\theta^\intercal X_t}$ is set, and revenue is realized. Through this lens, we can see that the Oracle is equivalent to pulling the arm $(z^*, \theta_0)$ at every $t$ in the new multi-armed bandit we have defined. DEEP-C is an arm-elimination algorithm for this multi-armed bandit.

From a learning standpoint, the goal is to learn the optimal $(z^*, \theta_0)$, which at the first sight seems to suffer from the curse of dimensionality. However, we observe that in fact, our problem allows for "free exploration" that lets us to learn efficiently in this setting; in particular, given $X_t$, for each choice of price $p_t$ we *simultaneously* obtain information about the expected revenue for a *range* of pairs $(z, \theta)$. This is specifically because we observe the context $X_t$, and because of the particular structure of demand that we consider. However, to ensure that each candidate $(z, \theta)$ arm has sufficiently high probability of being pulled at any time step, DEEP-C selects prices at random from a set of active prices, and ensures that this set is kept small via arm-elimination. The speedup in learning thus afforded enables us to obtain low regret.

Formally, our procedure is defined as follows. We partition the support of $Z_1$ into intervals of length $n^{-1/4}$. If the boundary sets are smaller, we enlarge the support slightly (by an amount less than $n^{-1/4}$) so that each interval is of equal length, and equal to $n^{-1/4}$. Let the corresponding intervals be $\mathcal{Z}_1, \ldots, \mathcal{Z}_k$, and their centroids be $\zeta_1, \ldots, \zeta_k$ where $k$ is less than or equal to $n^{1/4}$. Similarly, for $l = 1, 2, \ldots, d$, we partition the projection of the support of $\theta_0$ into the $l^{th}$ dimension into $k_l$ intervals of equal length $n^{-1/4}$, with sets $\Theta_1^{(l)}, \ldots, \Theta_{k_l}^{(l)}$ and centroids $\theta_1^{(l)}, \ldots, \theta_{k_l}^{(l)}$. Again, if the boundary sets are smaller, we enlarge the support so that each interval is of equal length $n^{-1/4}$.

Our algorithm keeps a set of active $(z, \theta) \subset \mathcal{Z} \times \Theta$ and eliminates those for which we have sufficient evidence for being far from $(z^*, \theta_0)$. We let $A(t) \subset \{1, \ldots, k\}^{d+1}$ represent a set of active cells, where a cell represents a tuple $(i, j_1, \ldots, j_d)$. Then, $\bigcup_{(i, j_1, \ldots, j_d) \in A(t)} \mathcal{Z}_i \times \prod_{i=1}^d \Theta_{j_l}^{(l)}$ represents the set of active $(z, \theta)$ pairs. Here, $A(1)$ contains all cells.

At each time $t$ we have a set of active prices, which depends on $X_t$ and $A(t)$, i.e.,

$$P(t) = \left\{ p : \exists (z, \theta) \in \bigcup_{(i, j_1, \ldots, j_d) \in A(t)} \mathcal{Z}_i \times \prod_{l=1}^{d} \Theta_{j_l}^{(l)} \text{ s.t. } \ln p = \ln z + \theta^{\mathsf{T}} X_t \right\}.$$

At time $t$ we pick a price $p_t$ from $P(t)$ uniformly at random. We say that cell $(i, j_1, \ldots, j_d)$ is *checked* if $p_t \in P_{i, j_1, \ldots, j_d}(t)$ where

$$P_{i, j_1, \ldots, j_d}(t) \triangleq \left\{ p : \exists z \in \mathcal{Z}_i, \exists \theta \in \prod_{l=1}^{d} \Theta_{j_l}^{(l)} \text{ s.t. } \ln p = \ln z + \theta^{\mathsf{T}} X_t \right\}.$$

Each price selection checks one or more cells $(i, j_1, \ldots, j_d)$.

Recall that the reward generated at time $t$ is $Y_t p_t$. Let $T_t(i, j_1, \ldots, j_d)$ be the number of times cell $(i, j_1, \ldots, j_d)$ is checked until time $t$, and let $S_t(i, j_1, \ldots, j_d)$ be the total reward obtained at these times. Let

$$\hat{\mu}_t(i, j_1, \ldots, j_d) = \frac{S_t(i, j_1, \ldots, j_d)}{T_t(i, j_1, \ldots, j_d)}.$$

We also compute confidence bounds for $\hat{\mu}_t(i, j_1, \ldots, j_d)$, as follows. Fix $\gamma > 0$. For each active $(i, j_1, \ldots, j_d)$, let

$$u_t(i, j_1, \ldots, j_d) = \hat{\mu}_t(i, j_1, \ldots, j_d) + \sqrt{\frac{\gamma}{T_t(i, j_1, \ldots, j_d)}},$$

and

$$l_t(i, j_1, \ldots, j_d) = \hat{\mu}_t(i, j_1, \ldots, j_d) - \sqrt{\frac{\gamma}{T_t(i, j_1, \ldots, j_d)}}.$$

These represent the upper and lower confidence bounds, respectively.

We eliminate $(i, j_1, \ldots, j_d) \in A(t)$ from $A(t+1)$ if there exists $(i', j_1', \ldots, j_d') \in A(t)$ such that

$$u_t(i, j_1, \ldots, j_d) < l_t(i', j_1', \ldots, j_d').$$

### 3.2 Variants of DEEP-C

*DEEP-C with Rounds:* Theoretical analysis of regret for arm elimination algorithms typically involves tracking the number of times each sub-optimal arm is pulled before being eliminated. However, this is challenging in our setting, since the set of arms which get "pulled" at an offered price depends on the covariate vector at that time. To resolve this challenge, we consider a variant where the algorithm operates in rounds, as follows.

Within a round the set of active sells remains unchanged. Further, we ensure that within each round each arm in the active set is pulled at least once. For our analysis, we keep track of only the first time an arm is pulled in each round, and ignore the rest. While this may seem wasteful, a surprising aspect of our analysis is that the regret cost incurred by this form of exploration is only poly-logarithmic in $n$. Further, since the number of times each arm is "explored" in each round is exactly one, theoretical analysis now becomes tractable. For formal definitions of this policy and also of the policies below, we refer the reader to Appendix B.

*Decoupled DEEP-C:* We now present a policy which has low computational complexity under sparsity and which does not suffer from price endogeneity, but may incur higher regret. At times $t = 1, 2, \ldots, \tau$, the price is set independently and uniformly at random from a compact set. This ensures that outputs $(Y_t : t = 1, 2, \ldots, \tau)$ are conditionally independent given covariates $(X_t : t = 1, 2, \ldots, \tau)$, i.e., there is no price endogeneity. We then use a low-complexity semi-parametric regression technique from Plan and Vershynin (2013) to estimate $\theta_0$ under a sparsity assumption. With estimation of $\theta_0$ in place, at times $t = \tau + 1, \ldots, n$, we use a one-dimensional version of DEEP-C to simultaneously estimate $z^*$ and maximize revenue. The best possible regret achievable with this policy is $\tilde{O}(n^{2/3})$, achieved when $\tau$ is $O(n^{2/3})$ Plan and Vershynin (2013).

*Sparse DEEP-C:* This policy also leverages sparsity, but without decoupling estimation of $\theta_0$ from estimation of $z^*$ and revenue maximization. At each time $t$, using the data collected in past we estimate $\theta_0$ via semi-perametric regression technique from Plan and Vershynin (2013). Using this estimate of $\theta_0$, the estimate of rewards for different values of $z$ from samples collected in past, and the corresponding confidence bounds, we obtain a set of active prices at each time, similar to that of DEEP-C, from which the price is picked at random.

While Sparse DEEP-C suffers from price endogeneity, with an appropriate choice of $\gamma$ we conjecture that its cost in terms of expected regret can be made poly-logarithmic in $n$; proving this result remains an important open direction. The intuition for this comes from our theoretical analysis of DEEP-C with Rounds and the following observation: even though the set of active prices may be different at different times, we still choose prices at random, and prices are eliminated only upon reception of sufficient evidence of suboptimality. We conjecture that these features are sufficient to ensure that the error in the estimate of $\theta_0$ is kept small with high probability. Our simulation results indeed show that this algorithm performs relatively well.

## 4  Regret analysis

The main theoretical result of this paper is the following. The regret bound below is achieved by DEEP-C with Rounds as defined in Section 3.2. For its proof see Appendix C.

**Theorem 1** *Under A1, A2, and A3, the expected regret under policy DEEP-C with Rounds with* $\gamma = \max\left(10\alpha_2^2, 4\frac{\kappa_2^2}{\log n}, \frac{\kappa_1^{-2}}{\log n}\right)$ *satisfies,*

$$\mathbb{E}[R_n] \leq 16000\alpha_1^{-2}\alpha_2^2\kappa_1^{-2}\kappa_2^{3/2}\gamma^{3/4}d^{11/4}n^{1/2}\log^{7/4} n + 5\alpha_2.$$

First, note that the above scaling is optimal w.r.t. $n$ (up to polylogarithmic factors), as even for the case where $X_t = 0$ w.p.1. it is known that achieving $o(\sqrt{n})$ expected regret is not possible (see Kleinberg and Leighton (2003)).

Second, we state our results with explicit dependence on various parameters discussed in our assumptions in order for the reader to track the ultimate dependence on the dimension $d$. Note that, as $d$ scales, the supports $\Theta$ and $\mathcal{X}$, and the distribution of $X$ may change. In turn, the parameters $\alpha_1$, $\alpha_2$, $\kappa_1$ and $\kappa_2$ which are constants for a given $d$, may scale as $d$ scales. These scalings need to be computed case by case as it depends on how one models the changes in $\Theta$ and $\mathcal{X}$. Below we discuss briefly how these may scale in practice.

Recall that $\alpha_1$ and $\alpha_2$ are bounds on $ze^{\theta^\intercal x}$, namely, the user valuations. Thus, it is meaningful to postulate that $\alpha_1$ and $\alpha_2$ do not scale with covariate dimension, as the role of covariates is to aid prediction of user valuations and not to change them. For example, one may postulate that $\theta_0$ is "sparse", i.e., the number of non-zero coordinates of $\theta_0$ is bounded from above by a known constant, in which case $\alpha_1$ and $\alpha_2$ do not scale with $d$. Dependence of $\kappa_1$ and $\kappa_2$ on $d$ is more subtle as they may depend on the details of the modeling assumptions. For example, their scaling may depend on scaling of the difference between the largest and second largest values of $r(z, \theta)$. One of the virtues of Theorem 1 is that it succinctly characterizes the scaling of regret via a small set of parameters.

Finally, the above result can be viewed through the lens of sample complexity. The arguments used in Lemma 1 and in the derivation of equation (4) imply that the sample complexity is "roughly" $O(\log(1/\delta)/\epsilon^2)$. More precisely, suppose that at a covariate vector $x$, we set the price $p(x)$. We say the mapping $p$ is *probably approximately revenue optimal* if for any $x$ the difference between the achieved revenue and the optimal revenue is at most $\epsilon$ with probability at least $1 - \delta$. The number of samples $m$ required to learn such a policy satisfies $m \operatorname{polylog}(m) \leq \frac{\log(1/\delta)}{\epsilon^2} f(d, \alpha_1, \alpha_2, \kappa_1, \kappa_2)$ where $f(\cdot)$ is polynomial function.

## 5  Simulation Results

**Simulation setup:** First, we simulate our model with covariate dimension $d = 2$, where covariate vectors are i.i.d. $d$-dimensional standard normal random vectors, the parameter space is $\Theta = [0, 1]^d$, the parameter vector is $\theta_0 = (1/\sqrt{2}, 1/\sqrt{2})$, the noise support is $\mathcal{Z} = [0, 1]$, and the noise distribution

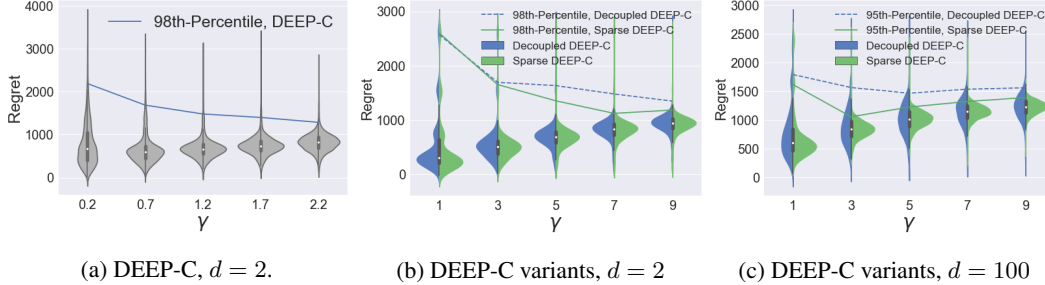

Figure 1: Regret comparison of the policies.

is $Z \sim \text{Uniform}([0,1])$. Note that even though we assumed that the covariate distribution has bounded support for ease of analysis, our policies do not assume that. Hence, we are able to use a covariate distribution with unbounded support in our simulations. In this setting, we simulate policies DEEP-C, Decoupled DEEP-C, and Sparse DEEP-C for time horizon $n = 10,000$ and for different values of parameter $\gamma$. Each policy is simulated 5,000 times for each set of parameters.

Next, we also simulate our model for $d = 100$ with $s = 4$ non-zero entries in $\theta_0$, with each non-zero entry equal to $1/\sqrt{s}$, each policy is simulated 1,500 times for each set of parameters, with the rest of the setup being the same as earlier. For this setup, we only simulate Decoupled DEEP-C and Sparse DEEP-C, as the computational complexity of DEEP-C does not scale well with $d$.

**Main findings:** First, we find that the performance of each policy is sensitive to the choice of $\gamma$, and that the range of $\gamma$ where expected regret is low may be different for different policies. The expected regret typically increases with increase in $\gamma$, however its variability typically reduces with $\gamma$. This is similar to the usual bias-variance tradeoff in learning problems. For our setup with $d = 2$, the reward of Oracle concentrates at around 4,150. As Figure 1 shows, each policy performs well in the plotted range of $\gamma$.

We find that the main metric where the performance of the policies is differentiated is in fact high quantiles of the regret distribution. For example, while the expected regret of DEEP-C at $\gamma = 2.2$ and that of Decoupled DEEP-C and Sparse DEEP-C at $\gamma = 7$ each are all roughly the same, the 98th-percentile of regret distribution under DEEP-C and Sparse DEEP-C is 13% and 24% lower than that under Decoupled DEEP-C, respectively.

For our setup with $d = 100$, while both Decoupled DEEP-C and Sparse DEEP-C perform similar in average regret, we find that Sparse DEEP-C significantly outperforms Decoupled DEEP-C in standard deviation and in 95th-percentile. In particular, 95th-percentile of Sparse DEEP-C is 33% lower than that under Decoupled DEEP-C.

# 6 Acknowledgments

This work was supported in part by National Science Foundation Grants DMS-1820942, DMS-1838576, CNS-1544548, and CNS-1343253. Any opinions, findings, and conclusions or recommendations expressed in this material are those of the authors and do not necessarily reflect the views of the National Science Foundation. We would like to thank Linjia Wu for reading and checking our proofs.

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
