[Supplementary Material · Camera_appendix.pdf]

# A  A class of examples where assumptions A1, A2, and A3 are satisfied

First consider a spherically distributed $d$ dimensional random vector $S$, i.e., for each $d$ dimensional orthonormal matrix $O$ the distributions of $S$ and $OS$ are identical. It is known that a $d$ dimensional random vector $S$ is spherically distributed iff there exists a positive (one dimensional) random variable $R$, called generating random variable, such that $S =_d R U^{(d)}$ where $U^{(d)}$ is uniformly distributed on the $d$ dimensional unit hypersphere Frahm (2004). For example, if $S$ is a standard normal random vector than $R^2$ is a chi-squared distributed random variable. Further, it is also known that for each spherically distributed $S$ there exists a function $\phi_S(.)$ such that the MGF of $S$, namely $\mathbb{E}[e^{\theta^\intercal S}]$, is equal to $\phi_S(\|\theta\|_2^2)$, where $\|.\|_2$ represents 2-norm Frahm (2004).

Now, suppose that $\{X_t\}_t$ are i.i.d. with a spherical distribution such that the generating random variable has density with support in $[0, \frac{1}{2}]$. Further suppose that $\{Z_t\}_t$ are i.i.d. Uniform$[0, 1)$, and that $\Theta \subset [0, 1]^d$. Thus A1 and A2 readily hold.

The following facts are easy to show: (i) $F(z) = z(1 - z)$ (ii) $z^* = 0.5$, (iii) $r(z, \theta) = z\phi_{X_1}(\|\theta\|_2^2) - z^2\phi_{X_1}(\|2\theta - \theta_0\|_2^2)$, and (iv) $(z^*, \theta_0)$ is the unique optimizer of $r(z, \theta)$. Further, $\phi_S(.)$ is a linear combination of MGFs Frahm (2004) which are convex, and is thus convex itself. Now, let $H$ be the Hessian of $r(z, \theta)$ at $(z^*, \theta_0)$. With some calculations one can show that for any non-zero $y = (z, \theta)$, we have that

$$y^\intercal H y = -4\phi_{X_1}''(\|\theta\|_2^2)\left(4z^2 + 4z\sum_{l=1}^{d}\theta^{(l)}\theta_0^{(l)} + 2\sum_{l=1}^{d}\sum_{l'=1}^{d}\theta^{(l)}\theta_0^{(l)}\theta^{(l')}\theta_0^{(l')} + \sum_{l=1}^{d}\left(\theta^{(l)}(1 + \theta_0^{(l)})\right)^2\right)$$

$$\leq -4\phi_{X_1}''(\|\theta\|_2^2)\left(2z^2 + 2(\theta^\intercal\theta_0 + z)^2 + \left(\sum_{l=1}^{d}|\theta^{(l)}|(1 + \theta_0^{(l)})\right)^2\right)$$

$$< 0$$

Thus, the Hessian of $r(z, \theta)$ at $(z^*, \theta_0)$ is negative definite. Thus, as argued in Section 2.3, A3 holds.

# B  Variants of DEEP-C: Formal Definitions

## B.1  DEEP-C with Rounds

We partition the support of $Z_1$ into intervals of length $n^{-1/4}$. If the boundary sets are smaller, we enlarge the support slightly (by an amount less than $n^{-1/4}$) so that each interval is of equal length, and equal to $n^{-1/4}$. Let the corresponding intervals be $\mathcal{Z}_1, \ldots, \mathcal{Z}_k$, and their centroids be $\zeta_1, \ldots, \zeta_k$ where $k$ is less than or equal to $n^{1/4}$. Similarly, for $l = 1, 2, \ldots, d$, we partition the projection of the support of the $\theta_0$ into the $l^{th}$ dimension into $k_l$ intervals of equal length, with sets $\Theta_1^{(l)}, \ldots, \Theta_{k_l}^{(l)}$ and centroids $\theta_1^{(l)}, \ldots, \theta_{k_l}^{(l)}$. Again, if the boundary sets are smaller, we enlarge the support so that each interval is of equal length, and equal to $n^{-1/4}$.

Our algorithm keeps a set of active $(z, \theta) \subset \mathcal{Z} \times \Theta$ and eliminates those for which we have sufficient evidence for being far from $(z^*, \theta_0)$.

Our algorithm operates in rounds. We use $\tau$ to index the round. Each round lasts for one or more time steps. Let $A(\tau) \subset \{1, \ldots, k\}$ where $\cup_{i \in A(\tau)}\mathcal{Z}_i$ represents the set of active $z$'s. For each $l$ let $B_l(\tau) \subset \{1, \ldots, k\}$ where $\prod_l \cup_{j \in B_l(\tau)}\Theta_j^{(l)}$ represents the set of active $\theta$'s in round $\tau$. Then, $(\cup_{i \in A(\tau)}\mathcal{Z}_i) \times \prod_l \cup_{j \in B_l(\tau)}\Theta_j^{(l)}$ represents the set of active $(z, \theta)$'s.

During each time $t$ in round $\tau$ we have a set of active prices, which depends on $X_t$ and $A(\tau) \times \prod_l B_l(\tau)$. Let

$$P(\tau, t) = \left\{p : \exists z \in \cup_{i \in A(\tau)}\mathcal{Z}_i, \exists \theta \in \prod_l \cup_{j \in B_l(\tau)}\Theta_j^{(l)} \text{ s.t. } \ln p = \ln z + \theta^\intercal X_t\right\}.$$

During round $\tau$, at each time $t$ we pick a price $p_t$ from $P(\tau, t)$ uniformly at random. At time $t$, we say that cell $(i, j_1, \ldots, j_d)$, i.e. set $\mathcal{Z}_i \times \Theta_{j_1}^{(1)} \times \Theta_{j_2}^{(2)} \times \ldots \times \Theta_{j_d}^{(d)}$, is 'checked' if $p_t \in P_{i,j_1,\ldots,j_d}(\tau, t)$ where

$$P_{i,j_1,\ldots,j_d}(\tau, t) \triangleq \left\{ p : \exists z \in \mathcal{Z}_i, \exists \theta \in \prod_l \Theta_{j_l}^{(l)} \text{ s.t. } \ln p = \ln z + \theta^\mathsf{T} X_t \right\}.$$

Each price selection checks one or more cells $(i, j_1, \ldots, j_d)$. The round lasts until all active cells are checked.

Let $t_\tau(i, j_1, \ldots, j_d)$ be the first time in round $\tau$ when the cell $(i, j_1, \ldots, j_d)$ is checked. Recall that the reward generated ay time $t$ is $Y_t p_t$. At the end of each round $\tau$, for each active cell $(i, j_1, \ldots, j_d)$ we compute the empirical average of the rewards generated at the times $t_{\tau'}(i, j_1, \ldots, j_d)$ for $\tau' = 1, \ldots, \tau$, i.e., we compute

$$\hat{\mu}_\tau(i, j_1, \ldots, j_d) = \frac{1}{\tau} \sum_{\tau'=1}^{\tau} Y_{t_{\tau'}(i,j_1,\ldots,j_d)} p_{t_{\tau'}(i,j_1,\ldots,j_d)}.$$

Note that for each cell, in each round we only record reward at the first time the cell is checked and ignore rewards at the rest of the times in that round. We also compute confidence bounds for $\hat{\mu}_\tau(i, j_1, \ldots, j_d)$, as follows. Let $\gamma = \max\left(10\alpha_2^2, 4\frac{\kappa_2^2}{\log n}, \frac{\kappa_1^{-2}}{\log n}\right)$. For each active $(i, j_1, \ldots, j_d)$, let

$$u_\tau(i, j_1, \ldots, j_d) = \hat{\mu}_\tau(i, j_1, \ldots, j_d) + \sqrt{\frac{\gamma d \log n}{\tau}},$$

and

$$l_\tau(i, j_1, \ldots, j_d) = \hat{\mu}_\tau(i, j_1, \ldots, j_d) - \sqrt{\frac{\gamma d \log n}{\tau}}.$$

These represent the upper and lower confidence bounds, respectively.

We eliminate $i \in A(\tau)$ from $A(\tau + 1)$ if there exists $i' \in A(\tau)$ such that

$$\sup_{(j_1,\ldots,j_d)\in\prod_l B_l(\tau)} u_\tau(i, j_1, \ldots, j_d) < \inf_{(j_1,\ldots,j_d)\in\prod_l B_l(\tau)} l_\tau(i', j_1, \ldots, j_d)$$

Similarly, we eliminate $j \in B_l(\tau)$ from $B_l(\tau + 1)$ if there exists $j' \in B_l(\tau)$ such that

$$\sup_{i\in A(\tau)} \sup_{(j_1,\ldots,j_{l-1},j_{l+1},\ldots,j_d)\in\prod_{l'\neq l} B_{l'}(\tau)} u_\tau(i, j_1, \ldots, j_{l-1}, j, j_{l+1}, \ldots, j_d)$$

$$< \inf_{i\in A(\tau)} \inf_{(j_1,\ldots,j_{l-1},j_{l+1},\ldots,j_d)\in\prod_{l'\neq l} B_{l'}(\tau)} l_\tau(i, j_1, \ldots, j_{l-1}, j', j_{l+1}, \ldots, j_d).$$

The time-complexity of this policy is driven by the number of cells, which increases as $O(n^{d/4})$, and thus scales poorly with $d$.

## B.2 Decoupled DEEP-C

We assume that there exists an $s \leq d$ such that at most $s$ entries in $\theta_0$ are non-zero. The value of $s$ is known to the platform. Here, $s$ represents sparsity and could be significantly smaller than $d$. We also assume that $\Theta \subset \{\theta : \|\theta\|_2 = 1\}$.

At times $t = 1, 2, \ldots, \lceil n \rceil^{2/3}$, select price uniformly at random from $[\alpha_1, \alpha_2]$. Then, we estimate $\theta_0$ by solving the following convex-optimization problem:

$$\begin{aligned} \underset{\theta}{\text{maximize}} \quad & \sum_{t=1}^{\lceil n \rceil^{2/3}} (2Y_t - 1)(\theta^\mathsf{T} X_t) \\ \text{subject to} \quad & \|\theta\|_1 \leq \frac{1}{\sqrt{s}}, \|\theta\|_2 \leq 1 \end{aligned} \tag{2}$$

We denote the estimate at $\hat{\theta}_0$.

We partition the support of $Z_1$ into intervals of length $n^{-1/4}$ as above, and let the corresponding intervals be $\mathcal{Z}_1, \ldots, \mathcal{Z}_k$ with centroids $\zeta_1, \ldots, \zeta_k$.

Fix $\gamma > 0$. For $t > \lceil n \rceil^{2/3}$ we do the following.

We let $A(t) \subset \{1, \ldots, k\}$ represent the set of active cells. Then, $\cup_{i \in A(t)} \mathcal{Z}_i$ represents the set of active $z$'s. Here, $A(\lceil n \rceil^{2/3} + 1) = \{1, \ldots, k\}$.

We let

$$P(t) = \left\{ p : \exists z \in \cup_{i \in A(t)} \mathcal{Z}_i \text{ s.t. } \ln p = \ln z + \hat{\theta}_0^\mathsf{T} X_t \right\}.$$

At time time $t$ we pick a price $p_t$ from $P(t)$ uniformly at random. We say that cell $i$, i.e. set $\mathcal{Z}_i$, is 'checked' if $p_t \in P_i(t)$ where

$$P_i(t) \triangleq \left\{ p : \exists z \in \mathcal{Z}_i \text{ s.t. } \ln p = \ln z + \hat{\theta}_0^\mathsf{T} X_t \right\}.$$

Each price selection checks one or more cells $i$. Let $T_t(i)$ be the number of times cell $i$ is checked till time $t$ and $S_t(i)$ be the total reward obtained at such times. Let

$$\hat{\mu}_t(i) = \frac{S_t(i)}{T_t(i)}.$$

We also compute confidence bounds for $\hat{\mu}_t(i)$, as follows. For each active $i$, let

$$u_t(i) = \hat{\mu}_t(i) + \sqrt{\frac{\gamma}{T_t(i)}},$$

and

$$l_t(i) = \hat{\mu}_t(i) - \sqrt{\frac{\gamma}{T_t(i)}}.$$

These represent the upper and lower confidence bounds, respectively.

We eliminate $i \in A(t)$ from $A(t+1)$ if there exists $i' \in A(t)$ such that

$$u_t(i) < l_t(i').$$

The time-complexity of this policy is driven by that of the convex-optimization problem (2), size of which scales as $O(n^{2/3}d)$. Note also that the total number of cells in this policy is $O(n^{1/4})$.

## B.3 Sparse DEEP-C

Again, we assume that there exists an $s \leq d$ such that at most $s$ entries in $\theta_0$ are non-zero, and that the value of $s$ is known to the platform. We also assume that $\Theta \subset \{\theta : \|\theta\|_2 = 1\}$.

We partition the support of $Z_1$ into intervals of length $n^{-1/4}$ as above, and let the corresponding intervals be $\mathcal{Z}_1, \ldots, \mathcal{Z}_k$ with centroids $\zeta_1, \ldots, \zeta_k$. We let $A(t) \subset \{1, \ldots, k\}$ represent a set of active cells at time $t$. Here, $A(1) = \{1, \ldots, k\}$. Fix $\gamma > 0$.

At each time $t$, estimate $\theta_0$ by solving the following convex-optimization problem:

$$\begin{aligned} \underset{\theta}{\text{maximize}} \quad & \sum_{t'=1}^{t-1} (2Y_{t'} - 1)(\theta^\mathsf{T} X_{t'}) \\ \text{subject to} \quad & \|\theta\|_1 \leq \frac{1}{\sqrt{s}}, \|\theta\|_2 \leq 1 \end{aligned} \tag{3}$$

We denote the estimate as $\hat{\theta}_0(t)$.

We let

$$P(t) = \left\{ p : \exists z \in \cup_{i \in A(t)} \mathcal{Z}_i \text{ s.t. } \ln p = \ln z + \hat{\theta}_0(t)^\intercal X_t \right\}.$$

At time time $t$ we pick a price $p_t$ from $P(t)$ uniformly at random. We say that cell $i$, i.e. set $\mathcal{Z}_i$, is 'checked' if $p_t \in P_i(t)$ where

$$P_i(t) \triangleq \left\{ p : \exists z \in \mathcal{Z}_i \text{ s.t. } \ln p = \ln z + \hat{\theta}_0(t)^\intercal X_t \right\}.$$

Each price selection checks one or more cells $i$. Let $T_t(i)$ be the number of times cell $i$ is checked till time $t$ and $S_t(i)$ be the total reward obtained at such times. Let

$$\hat{\mu}_t(i) = \frac{S_t(i)}{T_t(i)}.$$

We also compute confidence bounds for $\hat{\mu}_t(i)$, as follows. For each active $i$, let

$$u_t(i) = \hat{\mu}_t(i) + \sqrt{\frac{\gamma}{T_t(i)}},$$

and

$$l_t(i) = \hat{\mu}_t(i) - \sqrt{\frac{\gamma}{T_t(i)}}.$$

These represent the upper and lower confidence bounds, respectively.

We eliminate $i \in A(t)$ from $A(t+1)$ if there exists $i' \in A(t)$ such that

$$u_t(i) < l_t(i').$$

The time-complexity of this policy is driven by having to solve the convex-optimization problem (3) at each time $t$, size of which scales as $O(td)$. Its implementation at time $t$ can be sped up by using solution from time $t-1$ for initialization. Note also that the total number of cells in this policy is $O(n^{1/4})$.

## C  Proof of Theorem 1

Consider policy DEEP-C with Rounds as defined in Appendix B. The proof follows from a few technical results that we state now. We provide the statements of these results and delegate their proofs to Appendix D to not interrupt the logical flow of the proof of the theorem.

First, at the end of round $\tau$, with high probability, the set of active arms corresponds to cells with guaranteed $O\left(\sqrt{\frac{\log n}{\tau}}\right)$ expected regret. More precisely, recall the definitions of $r(z, \theta)$, $\zeta_i$, and $\theta_j^{(l)}$. Let

$$\Delta(i, j_1, \ldots, j_d) = r(z^*, \theta_0) - r\left(\zeta_i, (\theta_{j_l}^{(l)} : 1 \le l \le d)\right).$$

We have the following result.

**Lemma 1** *For each round $\tau$, let $E_1(\tau)$ be the event that the following holds:*

$$A(\tau) \subset \left\{ i : \sup_{(j_1, \ldots, j_d)} \Delta(i, j_1, \ldots, j_d) < 16\kappa_2 \kappa_1^{-1} \sqrt{\frac{\gamma d \log n}{\tau}} \right\},$$

*and for each $l$*

$$B_l(\tau) \subset \left\{ j : \sup_i \sup_{(j_1, \ldots, j_{l-1}, j_{l+1}, \ldots, j_d)} \Delta(i, j_1, \ldots, j_{l-1}, j, j_{l+1}, \ldots, j_d) < 16\kappa_2 \kappa_1^{-1} \sqrt{\frac{\gamma d \log n}{\tau}} \right\}.$$

*Then,*

$$\mathbb{P}(E_1(\tau)) \ge 1 - \frac{4}{n^2}.$$

Second, not only are the corresponding active cells guaranteed to have small expected regret with high probability, but the size (Lebesgue measure) of the set of active prices is guaranteed to be small with high probability. The next result provides explicit bound on such size.

**Lemma 2** *For each $\tau$, the event $E_1(\tau)$ implies that the following holds for each time $t$ in round $\tau$:*

$$L(P(\tau,t)) \leq 40 \frac{\alpha_2^2}{\alpha_1} d\kappa_1^{-1}\kappa_2^{1/2} \left(\frac{\gamma d \log n}{\tau}\right)^{1/4},$$

*where for each Borel set $A$, $L(A)$ is its Lebesgue measure.*

Third, after verifying that the remaining cells have a suitably controlled expected regret, and that the size of active arms (prices) is also controlled, we verify that at each time in the current round any given active cell is checked with substantially high probability.

**Lemma 3** *Fix round $\tau$. Consider an active cell $(i, j_1, \ldots, j_d)$. Then the probability that the cell $(i, j_1, \ldots, j_d)$ is checked at time $t$ in round $\tau$ is at least $\frac{\alpha_1 n^{-1/4}}{L(P(\tau,t))}$.*

Finally, using Lemmas 1, 2, and 3, we are ready to piece together all of the elements (i.e., control on the performance of active arms, size of the remaining arms, and the speed at which arms are explored) to obtain the main result, as we do next.

From Lemma 2 we have w.p. 1 that $L(P(\tau,t)) \leq \delta' \triangleq 40\frac{\alpha_2^2}{\alpha_1}d\kappa_1^{-1}\kappa_2^{1/2}\left(\frac{\gamma d \log n}{\tau}\right)^{1/4}$ for each $\tau$ and $t$.

Let $E_2(\tau)$ be the event that the round $\tau$ runs for at most $\frac{3d\delta'}{\alpha_1 n^{-1/4}}\log n$ times. Since the number of cells is at most $n^{d/4}$, by Lemma 3 and union bound we obtain:

$$\mathbb{P}((E_2(\tau))^c) \leq n^{d/4}\left(1 - \frac{\alpha_1 n^{-1/4}}{\delta'}\right)^{3d\frac{\delta'}{\alpha_1 n^{-1/4}}\log n} \leq n^{d/4}e^{-3d\log n} \leq n^{d/4-3d} \leq n^{-2d}$$

$$\leq n^{-2} \quad (4)$$

Also, recall event $E_1(\tau)$ from Lemma 1. By the law of total expectation, the expected regret incurred during round $\tau$, i.e. the difference between expected reward earned by the oracle and the platform during round $\tau$, denoted as $\tilde{R}_\tau$, satisfies the following:

$$\mathbb{E}[\tilde{R}_\tau] \leq \mathbb{E}[\tilde{R}_\tau|E_1(\tau), E_2(\tau)]P(E_1(\tau)\cap E_2(\tau)) + \mathbb{E}[\tilde{R}_\tau|E_1(\tau)^c\cup E_2(\tau)^c]\mathbb{P}(E_1(\tau)^c\cup E_2(\tau)^c).$$

Here, $P(E_1(\tau) \cap E_2(\tau)) \leq 1$, and $\mathbb{E}[\tilde{R}_\tau|E_2(\tau)^c \cup E_1(\tau)^c] \leq \alpha_2 n$ since the reward by the Oracle at any time $t$ is $z^* e^{\theta_0 X_t}\mathbf{1}\{V_t \geq p_t\} \leq z^* e^{\theta_0 X_t} \leq z^* \alpha_2 \leq \alpha_2$, with probability 1. Thus,

$$\mathbb{E}[\tilde{R}_\tau] \leq \mathbb{E}[\tilde{R}_\tau|E_2(\tau), E_1(\tau)] + \alpha_2 n\mathbb{P}((E_1(\tau)^c \cup E_2(\tau)^c)$$
$$\leq \mathbb{E}[\tilde{R}_\tau|E_2(\tau), E_1(\tau)] + \alpha_2 n \left(\mathbb{P}((E_1(\tau)^c) + \mathbb{P}((E_2(\tau)^c)\right)$$

Further, from (4) we have that $\mathbb{P}((E_2(\tau)^c) \leq n^{-2}$, and from Lemma 1 we have that $\mathbb{P}((E_1(\tau)^c) \leq 4n^{-2}$. Also, conditioned on events $E_1(\tau)$ and $E_2(\tau)$, we have the following:

(1) each round $\tau$ is of length at most $3d\log n \frac{\delta'}{\alpha_1 n^{-1/4}}$ (form the definition of $E_2(\tau)$), and

(2) the regret incurred is at most $16\kappa_2\kappa_1^{-1}\sqrt{\frac{\gamma d \log n}{\tau}}$ (from the definition of $E_1(\tau)$),

(3) $\delta' = 40\frac{\alpha_2^2}{\alpha_1}d\kappa_1^{-1}\kappa_2^{1/2}\left(\frac{\gamma d \log n}{\tau}\right)^{1/4}$ (from definition of $\delta'$).

Thus, we get

$$\mathbb{E}[\tilde{R}_\tau] \leq \left(3d\log n \frac{40\frac{\alpha_2^2}{\alpha_1}d\kappa_1^{-1}\kappa_2^{1/2}\left(\frac{\gamma d\log n}{\tau}\right)^{1/4}}{\alpha_1 n^{-1/4}}\right)\left(16\kappa_2\kappa_1^{-1}\sqrt{\frac{\gamma d\log n}{\tau}}\right) + \frac{5\alpha_2}{n}.$$

Upon simplification, we obtain

$$\mathbb{E}[\tilde{R}_\tau] \leq 1920\alpha_2^2\alpha_1^{-2}\kappa_1^{-2}\kappa_2^{3/2}\gamma^{3/4}d^{11/4}n^{1/4}\log^{7/4}n\tau^{-3/4} + \frac{5\alpha_2}{n}.$$

Thus, the total expected regret satisfies:

$$\mathbb{E}[R_n] \leq \sum_{\tau=1}^{n}\tilde{R}_\tau \leq 2000\alpha_2^2\alpha_1^{-2}\kappa_1^{-2}\kappa_2^{3/2}\gamma^{3/4}d^{11/4}n^{1/4}\log^{7/4}n\sum_{\tau=1}^{n}\tau^{-3/4} + 5\alpha_2$$

$$\leq 16000\alpha_1^{-2}\alpha_2^2\kappa_1^{-2}\kappa_2^{3/2}\gamma^{3/4}d^{11/4}n^{1/2}\log^{7/4}n + 5\alpha_2.$$

Hence, the theorem holds. ∎

## D Proof of lemmas used in Theorem 1

We present the proofs of Lemmas 1, 2, and 3 in order.

**Proof of Lemma 1:** For notational convenience and simplification of regret analysis, we pretend that the following happens at the end of a round: We simulate 'virtual times' during which we obtain virtual covariates and virtual prices so that we obtain a sample for each inactive set as well at round $\tau$, and update $u_\tau$ and $l_\tau$ accordingly. These times do not count as real times, and since inactive sets do not take part in any decision making, the above procedure at virtual times incur no cost and have no bearing to the execution of the actual algorithm in practice.

Throughout our development, we shall use that, as stated in A3,

$$\kappa_1 \max\left\{(z^*-\zeta_i)^2, \max_{1\leq l\leq d}(\theta_0^{(l)}-\theta_{j_l}^{(l)})^2\right\} \leq \Delta(i,j_1,\ldots,j_d) \leq \frac{\kappa_2}{d+1}\|(z^*-\zeta_i,\theta_0-(\theta_{j_l}^{(l)}:1\leq l\leq d))\|^2$$

Fix a cell $(i,j_1,\ldots,j_d)$ such that $\Delta(i,j_1,\ldots,j_d) > 16\kappa_2\kappa_1^{-1}\sqrt{\frac{\gamma d\log n}{\tau}}$. If no such cell exists, then there is is nothing to prove since in that case $\mathbb{P}(E_1(\tau)) = 1$. We show that the probability of such a cell being eliminated is high. Let $E'$ be the event that cell $(i,j_1,\ldots,j_d)$ has not been eliminated by the end of round $\tau$. In addition, let $E_m^*$ be the event that $(i^*,j_1^*,\ldots,j_d^*)$ is eliminated at round $m$, where $(i^*,j_1^*,\ldots,j_d^*)$ is the cell that contains $(z^*,\theta_0)$. Using union bound, we can write

$$\mathbb{P}(E') = \mathbb{P}\left(E'\cap(\cup_{m=1}^{\tau}E_m^*)\right) + \mathbb{P}\left(E'\cap(\cap_{m=1}^{\tau}(E_m^*)^c)\right)$$

$$\leq \sum_{m=1}^{\tau}\mathbb{P}(E_m^*) + \mathbb{P}\left(E'\cap(\cap_{m=1}^{\tau}(E_m^*)^c)\right).$$

We have two claims,

**Claim 1:** $\mathbb{P}(E_m^*) \leq 2\frac{1}{n^{4d}}$, and

**Claim 2:** $\mathbb{P}\left(E'\cap(\cap_{m=1}^{\tau}(E_m^*)^c)\right) \leq \frac{2}{n^{10d}}$.

It follows directly from Claims 1 and 2, since and $\tau \leq n$, that

$$\mathbb{P}(E') \leq \tau\frac{2}{n^{4d}} + \frac{2}{n^{10d}} \leq \frac{1}{n^{3d}} + \frac{2}{n^{10d}} \leq \frac{4}{n^{3d}}.$$

Since total number of cells is at most $n^{d/4}$, we have that

$$\mathbb{P}((E_1(\tau))^c) \le n^{d/4} \frac{4}{n^{3d}} \le \frac{4}{n^{11d/4}} \le \frac{4}{n^{11/4}},$$

and hence the lemma would follow. So, we just need to establish Claim 1 and Claim 2.

For Claim 1, note that

$$
\begin{aligned}
\mathbb{P}(E_m^*) \le & \mathbb{P}\left(\exists (i, j_1, \ldots, j_d) \text{ s.t. } u_\tau(i^*, j_1^*, \ldots, j_d^*) < l_\tau(i, j_1, \ldots, j_d)\right) \\
\le & n^{d/4} \sup_{(i,j_1,\ldots,j_d)} \mathbb{P}\left(u_\tau(i^*, j_1^*, \ldots, j_d^*) < l_\tau(i, j_1, \ldots, j_d)\right) \\
\le & n^{d/4} \sup_{(i,j_1,\ldots,j_d)} \left( \mathbb{P}\left(u_\tau(i^*, j_1^*, \ldots, j_d^*) < \inf_{(z,\theta) \in \mathcal{Z}_{i^*} \times \Theta_{j_1^*} \times \ldots \times \Theta_{j_d^*}} r(z,\theta)\right) \right. \\
& \left. + \mathbb{P}\left(l_\tau(i, j_1, \ldots, j_d) \ge \inf_{(z,\theta) \in \mathcal{Z}_{i^*} \times \Theta_{j_1^*} \times \ldots \times \Theta_{j_d^*}} r(z,\theta)\right)\right),
\end{aligned}
$$

where the last inequality follows from the fact that $l < u$ implies that for each $c$ we have $l < c$ or $u \ge c$; we are choosing $c = \inf_{(z,\theta) \in \mathcal{Z}_{i^*} \times \Theta_{j_1^*} \times \ldots \times \Theta_{j_d^*}} r(z,\theta)$. Further, we have

$$
\begin{aligned}
& \mathbb{P}\left(u_\tau(i^*, j_1^*, \ldots, j_d^*) \le \inf_{(z,\theta) \in \mathcal{Z}_{i^*} \times \Theta_{j_1^*} \times \ldots \times \Theta_{j_d^*}} r(z,\theta)\right) \\
= & \mathbb{P}\left(\hat{\mu}_\tau(i^*, j_1^*, \ldots, j_d^*) \le \inf_{(z,\theta) \in \mathcal{Z}_{i^*} \times \Theta_{j_1^*} \times \ldots \times \Theta_{j_d^*}} r(z,\theta) - \sqrt{\frac{\gamma d \log n}{\tau}}\right) \\
\le & \mathbb{P}\left(\hat{\mu}_\tau(i^*, j_1^*, \ldots, j_d^*) \le \mathbb{E}[\hat{\mu}_\tau(i^*, j_1^*, \ldots, j_d^*)] - \sqrt{\frac{\gamma d \log n}{\tau}}\right)
\end{aligned}
$$

Note that

$$0 \le \hat{\mu}_\tau(i^*, j_1^*, \ldots, j_d^*) \le \sup_{x \in \mathcal{X}, z \in \mathcal{Z}, \theta \in \Theta} z e^{\theta^\top x} \le \alpha_2.$$

Thus, using Hoeffding's inequality, we obtain

$$
\begin{aligned}
\mathbb{P}\left(u_\tau(i^*, j_1^*, \ldots, j_d^*) \le \inf_{(z,\theta) \in \mathcal{Z}_{i^*} \times \Theta_{j_1^*} \times \ldots \times \Theta_{j_d^*}} r(z,\theta)\right) & \le e^{-\frac{2\gamma d \log n}{\alpha_2^2}} \\
& \le e^{-20d \log n} \\
& \le \frac{1}{n^{20d}}.
\end{aligned}
$$

Fix $(i, j_1, \ldots, j_d)$. From A3 and the fact that each cell is of size $n^{-1/4}$, we have $r(z^*, \theta_0) - \inf_{(z,\theta) \in \mathcal{Z}_{i^*} \times \Theta_{j_1^*} \times \ldots \times \Theta_{j_d^*}} r(z,\theta) \le \kappa_2 (n^{-1/4})^2$. Also, from the definition of $\gamma$ we have that $\kappa_2 \le \sqrt{\frac{\gamma d \log n}{4}}$. Since $\tau \le n$ we get $\kappa_2 (n^{-1/4})^2 \le \sqrt{\frac{\gamma d \log n}{4\tau}}$.

Thus, we get that

$$\sup_{(z,\theta) \in \mathcal{Z}_i \times \Theta_{j_1}^{(1)} \times \ldots \times \Theta_{j_d}^{(d)}} r(z,\theta) \le r(z^*, \theta_0) \le \inf_{(z,\theta) \in \mathcal{Z}_{i^*} \times \Theta_{j_1^*} \times \ldots \times \Theta_{j_d^*}} r(z,\theta) + \sqrt{\frac{\gamma d \log n}{4\tau}}.$$

Thus,

$$\mathbb{P}\left(l_\tau(i, j_1, \ldots, j_d) \geq \inf_{(z,\theta) \in \mathcal{Z}_{i^*} \times \Theta_{j_1^*} \times \ldots \times \Theta_{j_d^*}} r(z, \theta)\right)$$

$$\leq \mathbb{P}\left(l_\tau(i, j_1, \ldots, j_d) \geq \sup_{(z,\theta) \in \mathcal{Z}_i \times \Theta_{j_1}^{(1)} \times \ldots \times \Theta_{j_d}^{(d)}} r(z, \theta) - \sqrt{\frac{\gamma d \log n}{4\tau}}\right)$$

$$= \mathbb{P}\left(\hat{\mu}_\tau(i, j_1, \ldots, j_d) \geq \sup_{(z,\theta) \in \mathcal{Z}_i \times \Theta_{j_1}^{(1)} \times \ldots \times \Theta_{j_d}^{(d)}} r(z, \theta) + \sqrt{\frac{\gamma d \log n}{4\tau}}\right)$$

$$\leq e^{-\frac{\gamma d \log n}{2\alpha_2^2}}$$

$$\leq e^{-5d \log n}$$

$$\leq \frac{1}{n^{5d}}$$

Thus,

$$\mathbb{P}(E_m^*) \leq 2\frac{1}{n^{5d - d/4}} \leq 2\frac{1}{n^{4d}}.$$

Hence, the Claim 1 follows. We now show Claim 2. Note that

$$\mathbb{P}\left(E' \cap (\cap_{m=1}^\tau (E_m^*)^c)\right) \leq \mathbb{P}\left(u_\tau(i, j_1, \ldots, j_d) \geq l_\tau(i^*, j_1^*, \ldots, j_d^*)\right).$$

Let $(z', \theta') \in \arg\sup_{(z,\theta) \in \mathcal{Z}_i \times \Theta_{j_1}^{(1)} \times \ldots \times \Theta_{j_d}^{(d)}} r(z, \theta)$. Using the fact that for any $u, l, c$ we have that $u \geq l$ implies $u \geq c$ or $c \geq l$, and letting $c = (r(z^*, \theta_0) - r(z', \theta'))/2$ we obtain

$$\mathbb{P}\left(u_\tau(i, j_1, \ldots, j_d) \geq l_\tau(i^*, j_1^*, \ldots, j_d^*)\right)$$
$$\leq \mathbb{P}\left(u_\tau(i, j_1, \ldots, j_d) \geq (r(z^*, \theta_0) - r(z', \theta'))/2 + r(z', \theta')\right)$$
$$+ \mathbb{P}\left(l_\tau(i^*, j_1^*, \ldots, j_d^*) \leq r(z^*, \theta_0) - (r(z^*, \theta_0) - r(z', \theta'))/2\right). \quad (5)$$

Now, by A3 and using the fact that $\gamma \geq \frac{\kappa_1^{-2}}{\log n}$, we obtain that

$$\|(z^* - \zeta_i, \theta_0 - (\theta_{j_k}^{(l)} : 1 \leq l \leq k)\|^2 \geq \kappa_2^{-1}(d+1)\Delta(i, j_1, \ldots, j_d) \geq 16\kappa_1^{-1}(d+1)\sqrt{\frac{\gamma d \log n}{\tau}}$$

$$\geq 16(d+1)\sqrt{\frac{d}{\tau}} \geq 16(d+1)\sqrt{\frac{1}{n}}.$$

Further, by construction of the partition, we have $|z' - \zeta_i| \leq \frac{1}{2}n^{-1/4}$ and $(\theta'^{(l)} - \theta_{j_l}^{(l)}) \leq \frac{1}{2}n^{-1/4}$ for each $1 \leq l \leq d$. Thus,

$$\left\|\left(z^* - \zeta_i, \theta_0 - (\theta_{j_l}^{(l)} : 1 \leq l \leq d)\right)\right\|^2 \leq \|(z^* - z', \theta_0 - \theta')\|^2 + \left\|\left(z' - \zeta_i, \theta' - (\theta_{j_l}^{(l)} : 1 \leq l \leq d)\right)\right\|^2$$

$$\leq \|(z^* - z', \theta_0 - \theta')\|^2 + (d+1)\left(\frac{n^{-1/4}}{2}\right)^2.$$

In turn, we have

$$\left\| (z^* - z', \theta_0 - \theta') \right\|^2 \geq \left\| \left( z^* - \zeta_i, \theta_0 - (\theta_{j_l}^{(l)} : 1 \leq l \leq d) \right) \right\|^2 - (d+1) \left( \frac{n^{-1/4}}{2} \right)^2.$$

Thus, by again using A3 we get

$$
\begin{aligned}
\frac{\Delta(i, j_1, \ldots, j_d)}{r(z^*, \theta_0) - r(z', \theta')} &\leq \frac{\kappa_2 \left\| \left( z^* - \zeta_i, \theta_0 - (\theta_{j_l}^{(l)} : 1 \leq l \leq d) \right) \right\|^2}{(d+1)\kappa_1 \max \left\{ (z^* - z)^2, \max_{1 \leq l \leq d} (\theta_0^{(\ell)} - \theta^{(l)})^2 \right\}} \\
&\leq \frac{\kappa_2 \left\| \left( z^* - \zeta_i, \theta_0 - (\theta_{j_l}^{(l)} : 1 \leq l \leq d) \right) \right\|^2}{\kappa_1 \left\| (z^* - z', \theta_0 - \theta') \right\|^2} \\
&\leq \kappa_2 \kappa_1^{-1} \left( 1 - \frac{(d+1) \left( \frac{n^{-1/4}}{2} \right)^2}{\left\| \left( z^* - \zeta_i, \theta_0 - (\theta_{j_l}^{(l)} : 1 \leq l \leq d) \right) \right\|^2} \right)^{-1} \\
&\leq \kappa_2 \kappa_1^{-1} (1 - \frac{1/4}{16})^{-1} \leq 4\kappa_2 \kappa_1^{-1}.
\end{aligned}
$$

Thus, we get

$$\Delta(i, j_1, \ldots, j_d) \leq 4\kappa_2 \kappa_1^{-1} (r(z^*, \theta_0) - r(z', \theta')). \tag{6}$$

Consequently,

$$
\begin{aligned}
&\mathbb{P}\left( u_\tau(i, j_1, \ldots, j_d) \geq \left( r(z^*, \theta_0) - r(z', \theta') \right) / 2 + r(z', \theta') \right) \\
&\leq \mathbb{P}\left( u_\tau(i, j_1, \ldots, j_d) \geq \Delta(i, j_1, \ldots, j_d)/(8\kappa_2\kappa_1^{-1}) + r(z', \theta') \right) \\
&\leq \mathbb{P}\left( u_\tau(i, j_1, \ldots, j_d) \geq 2\sqrt{\frac{\gamma d \log n}{\tau}} + \sup_{(z,\theta) \in \mathcal{Z}_i \times \Theta_{j_1}^{(1)} \times \ldots \times \Theta_{j_d}^{(d)}} r(z, \theta) \right) \\
&\leq \mathbb{P}\left( \hat{\mu}_\tau(i, j_1, \ldots, j_d) \geq 2\sqrt{\frac{\gamma d \log n}{\tau}} - \sqrt{\frac{\gamma d \log n}{\tau}} + \sup_{(z,\theta) \in \mathcal{Z}_i \times \Theta_{j_1}^{(1)} \times \ldots \times \Theta_{j_d}^{(d)}} r(z, \theta) \right) \\
&= \mathbb{P}\left( \hat{\mu}_\tau(i, j_1, \ldots, j_d) \geq \sqrt{\frac{\gamma d \log n}{\tau}} + \sup_{(z,\theta) \in \mathcal{Z}_i \times \Theta_{j_1}^{(1)} \times \ldots \times \Theta_{j_d}^{(d)}} r(z, \theta) \right) \\
&\leq \mathbb{P}\left( \hat{\mu}_\tau(i, j_1, \ldots, j_d) \geq \sqrt{\frac{\gamma d \log n}{\tau}} + \mathbb{E}[\hat{\mu}_\tau(i, j_1, \ldots, j_d)] \right)
\end{aligned}
$$

Again using Hoeffding's inequality, we get

$$\mathbb{P}\left( u_\tau(i, j_1, \ldots, j_d) \geq \left( r(z^*, \theta_0) - r(z', \theta') \right) / 2 + r(z', \theta') \right) \leq e^{-\frac{2\gamma d \log n}{\alpha_2^2}} \leq e^{-20 d \log n} \leq \frac{1}{n^{20d}}. \tag{7}$$

Now, recall that $r(z^*, \theta_0) - \inf_{(z,\theta) \in \mathcal{Z}_{i^*} \times \Theta_{j_1^*} \times \ldots \times \Theta_{j_d^*}} r(z, \theta) \leq \kappa_2 n^{-1/2} \leq \sqrt{\frac{\gamma d \log n}{\tau}}$. Thus, we have

$$\mathbb{P}\left(l_\tau(i^*, j_1^*, \ldots, j_d^*) \le r(z^*, \theta_0) - \left(r(z^*, \theta_0) - r((z', \theta'))\right)/2\right)$$

$$\le \mathbb{P}\left(l_\tau(i^*, j_1^*, \ldots, j_d^*) \le r(z^*, \theta_0) - \Delta(i, j_1, \ldots, j_d)/(8\kappa_2\kappa_1^{-1})\right)$$

$$\le \mathbb{P}\left(l_\tau(i^*, j_1^*, \ldots, j_d^*) \le r(z^*, \theta_0) - 2\sqrt{\frac{\gamma d \log n}{\tau}}\right)$$

$$\le \mathbb{P}\left(l_\tau(i^*, j_1^*, \ldots, j_d^*) \le \inf_{(z,\theta)\in\mathcal{Z}_{i^*}\times\Theta_{j_1^*}\times\ldots\times\Theta_{j_d^*}} r(z, \theta) + \sqrt{\frac{\gamma d \log n}{\tau}} - 2\sqrt{\frac{\gamma d \log n}{\tau}}\right)$$

$$\le \mathbb{P}\left(\hat{\mu}_\tau(i^*, j_1^*, \ldots, j_d^*) \le \inf_{(z,\theta)\in\mathcal{Z}_{i^*}\times\Theta_{j_1^*}\times\ldots\times\Theta_{j_d^*}} r(z, \theta) - \sqrt{\frac{\gamma d \log n}{\tau}}\right)$$

$$\le \mathbb{P}\left(\hat{\mu}_\tau(i^*, j_1^*, \ldots, j_d^*) \le \mathbb{E}[\hat{\mu}_\tau(i^*, j_1^*, \ldots, j_d^*)] - \sqrt{\frac{\gamma d \log n}{\tau}}\right)$$

Using Hoeffding's inequality yet again, we get

$$\mathbb{P}\left(l_\tau(i^*, j_1^*, \ldots, j_d^*) \le r(z^*, \theta_0) - \left(r(z^*, \theta_0) - r((z', \theta'))\right)/2\right)$$
$$\le e^{-\frac{2\gamma d \log n}{\alpha_2^2}} \le e^{-20d \log n} \le \frac{1}{n^{20d}}. \quad (8)$$

Claim 2 thus follows from (5), (7) and (8). This completes proof of Lemma 1. We now proceed with the proof of Lemma 2.

**Proof of Lemma 2:**

Note that, by translation invariance, $L(P(\tau, t)) = L\left(P(\tau, t) - z^* e^{\theta_0^\intercal x_t}\right)$. In addition, for any measurable set $A$, we always have the bound $L(A) \le 2\sum_{a\in A} |a|$. Therefore, by definition of $P(\tau, t)$, we have

$$L(P(\tau, t)) \le 2 \sup_{z\in\mathcal{Z}_A, \theta\in\Theta_A} |z^* e^{\theta_0^\intercal x_t} - z e^{\theta^\intercal x_t}|,$$

where $\mathcal{Z}_A$ and $\Theta_A$ be the set of active $z$ and $\theta$ in round $\tau$. Now, fix $(z, \theta) \in Z_A \times \Theta_a$. Let $z^* - z = \delta_z$ and $\theta_0 - \theta = \delta_\theta$. Then, at time $t$ in round $\tau$, we have

$$\begin{aligned}
z e^{\theta^\intercal x_t} &= (z^* - \delta_z) e^{\theta_0^\intercal x_t} e^{-\delta_\theta^\intercal x_t} \\
&= e^{\theta_0^\intercal x_t}(z^* - \delta_z)\left(1 - (1 - e^{-\delta_\theta^\intercal x_t})\right) \\
&= e^{\theta_0^\intercal x_t}\left(z^*\left(1 - (1 - e^{-\delta_\theta^\intercal x_t})\right) - \delta_z\left(1 - (1 - e^{-\delta_\theta^\intercal x_t})\right)\right) \\
&= e^{\theta_0^\intercal x_t}\left(z^* - z^*(1 - e^{-\delta_\theta^\intercal x_t}) - \delta_z + \delta_z(1 - e^{-\delta_\theta^\intercal x_t})\right) \\
&= e^{\theta_0^\intercal x_t} z^* + e^{\theta_0^\intercal x_t}\left(-z^*(1 - e^{-\delta_\theta^\intercal x_t}) - \delta_z e^{-\delta_\theta^\intercal x_t}\right) \\
&= e^{\theta_0^\intercal x_t} z^* - e^{\theta_0^\intercal x_t}\left(z^*(1 - e^{-\delta_\theta^\intercal x_t}) + \delta_z e^{-\delta_\theta^\intercal x_t}\right).
\end{aligned}$$

Recall that $\alpha_1 \le e^{\theta^\intercal x} \le \alpha_2$ for each $x \in \mathcal{X}$ and $\theta \in \Theta$. Thus,

$$e^{-\delta_\theta^\intercal x_t} = \frac{e^{\theta^\intercal x_t}}{e^{\theta_0^\intercal x_t}} \le \frac{\alpha_2}{\alpha_1}.$$

Thus, by triangle inequality, and noting that $z^* \le 1$ as $\mathcal{Z}$ is a subset of the unit interval, we have

$$L(P(\tau,t)) \leq 2 \sup_{z \in \mathcal{Z}_A, \theta \in \Theta_A} \left| e^{\theta_0^\intercal x_t} \left( z^*(1 - e^{-\delta_\theta^\intercal x_t}) + \delta_z e^{-\delta_\theta^\intercal x_t} \right) \right|$$

$$\leq 2\alpha_2 \left( \sup_{\theta \in \Theta_A} \left| z^*(1 - e^{-\delta_\theta^\intercal x_t}) \right| + \sup_{z \in \mathcal{Z}_A, \theta \in \Theta_A} \left| \delta_z e^{-\delta_\theta^\intercal x_t} \right| \right)$$

$$\leq 2\alpha_2 \sup_{\theta \in \Theta_A} \left| z^*(1 - e^{-\delta_\theta^\intercal x_t}) \right| + 2\alpha_2 \sup_{z \in \mathcal{Z}_A, \theta \in \Theta_A} \left| \delta_z e^{-\delta_\theta^\intercal x_t} \right|$$

$$\leq 2\alpha_2 \sup_{\theta \in \Theta_A} \left| (1 - e^{-\delta_\theta^\intercal x_t}) \right| + 2\frac{\alpha_2^2}{\alpha_1} \sup_{z \in \mathcal{Z}_A} |\delta_z|.$$

From Lemma 1, for each $\tau$ and each time $t$ in round $\tau$, with probability at least $1 - 4/n^2$ the only active cells $(i, j_1, \ldots, j_d)$ are the ones such that $\Delta(i, j_1, \ldots, j_d) \leq 16\kappa_2\kappa_1^{-1}\sqrt{\frac{\gamma d \log n}{\tau}}$. Thus, under $E_1(\tau)$, we have

$$\sup_{z \in \mathcal{Z}_A} \kappa_1 |\delta_z|^2 \leq 16\kappa_2\kappa_1^{-1}\sqrt{\frac{\gamma d \log n}{\tau}}.$$

Also, for each $\theta$,

$$\left| (1 - e^{-\delta_\theta^\intercal x_t}) \right| = \left| \delta_\theta^\intercal x_t - \frac{e^{-\delta}}{2}(\delta_\theta^\intercal x_t)^2 \right|,$$

for some $0 < |\delta| < |\delta_\theta^\intercal x_t|$. Since $0 < |\delta| < |\delta_\theta^\intercal x_t|$, we have $e^{-\delta} \leq \sup(1, e^{-\delta_\theta^\intercal x_t}) \leq \alpha_2/\alpha_1$. Thus, by triangle inequality and noting that $\mathcal{X}$ and $\Theta$ are a subset of unit hypercube, we get

$$\sup_{\theta \in \Theta_A} \left| (1 - e^{-\delta_\theta^\intercal x_t}) \right| \leq \sup_{\theta \in \Theta_A} |\delta_\theta^\intercal x_t| + \frac{\alpha_2}{\alpha_1} \sup_{\theta \in \Theta_A} |\delta_\theta^\intercal x_t|^2$$

$$\leq \sup_{\theta \in \Theta_A} \|\delta_\theta\|_1 \|x_t\|_\infty + \frac{\alpha_2}{\alpha_1} \sup_{\theta \in \Theta_A} \|\delta_\theta\|_1^2 \|x_t\|_\infty^2$$

$$\leq \sup_{\theta \in \Theta_A} \|\delta_\theta\|_1 + \frac{\alpha_2}{\alpha_1} \sup_{\theta \in \Theta_A} \|\delta_\theta\|_1^2$$

$$\leq 4\frac{\alpha_2}{\alpha_1} \sup_{\theta \in \Theta_A} \|\delta_\theta\|_1$$

$$\leq 4\frac{\alpha_2}{\alpha_1} d\kappa_1^{-1}\sqrt{16\kappa_2\sqrt{\frac{\gamma d \log n}{\tau}}}.$$

Thus, we get

$$\frac{1}{2}|P(\tau,t)| \leq \frac{\alpha_2^2}{\alpha_1}\kappa_1^{-1}\sqrt{16\kappa_2\kappa_1^{-1}\sqrt{\frac{\gamma d \log n}{\tau}}} + 4\frac{\alpha_2^2}{\alpha_1}d\kappa_1^{-1}\sqrt{16\kappa_2\sqrt{\frac{\gamma d \log n}{\tau}}},$$

$$\leq 5\frac{\alpha_2^2}{\alpha_1}d\kappa_1^{-1}\sqrt{16\kappa_2\sqrt{\frac{\gamma d \log n}{\tau}}}.$$

This completes the proof of Lemma 2. We now proceed to the proof of Lemma 3.

**Proof of Lemma 3:** Since the price at time $t$ is picked uniformly at random from $P(\tau,t)$, and since $P_{i,j_1,\ldots,j_d}(\tau,t) \subset P(\tau,t)$, we have that the probability that the cell $(i, j_1, \ldots, j_d)$ is checked at time $t$ in round $\tau$ is equal to $\frac{L(P_{i,j_1,\ldots,j_d}(\tau,t))}{L(P(\tau,t))}$. Thus, the result would follow if we show that $L(P_{i,j_1,\ldots,j_d}(\tau,t)) \geq n^{-1/4}\alpha_1$ w.p. 1. We show that below.

Fix $\theta$ from $\prod_l \Theta_{j_l}^{(l)}$. For each $x \in \mathcal{X}$ let

$$P(x) \triangleq \left\{ p : \exists z \in \mathcal{Z}_i \text{ s.t. } p = z e^{\theta^\mathsf{T} x} \right\}.$$

Since $L(\mathcal{Z}_i) = n^{-1/4}$, for each $x \in \mathcal{X}$ we have

$$L(P(x)) = n^{-1/4} e^{\theta^\mathsf{T} x} \geq n^{-1/4} \alpha_1.$$

Thus, $L(P(X_t)) \geq n^{-1/4} \alpha_1$ w.p. 1. But, by definition we have $P(X_t) \subset P_{i,j_1,\ldots,j_d}(\tau, t)$. Thus, $L(P_{i,j_1,\ldots,j_d}(\tau, t)) \geq n^{-1/4} \alpha_1$ w.p. 1. This completes the proof of Lemma 3.

# E  Extensions

## E.1  Incorporating adversarial covariates

We believe that the i.i.d. assumption on covariates can be significantly relaxed. As a prelude, consider the following modification to A1.

**A4** *We assume that $\{Z_t\}_t$ are i.i.d. with compact support $\mathcal{Z}$. We assume that the support of $X_t$ for each $t$ is compact, namely $\mathcal{X}$. Given the past, $X_t$ can be chosen adversarially from its support. More formally, we assume that $\mathcal{X}$ is $\sigma(X_1, \ldots, X_{t-1}, Z_1, \ldots, Z_{t-1}, p_1, \ldots, p_{t-1})$-measurable.*

Given Assumption A4, consider the following strengthening of Assumption A3. Recall that $F(z) = z\mathbb{P}(Z_1 > z)$. Let

$$r(z, \theta, x) = e^{\theta_0^\mathsf{T} x} F\left( e^{-(\theta_0 - \theta)^\mathsf{T} x} z \right).$$

Given covariate $x$, $r(z, \theta, x)$ can be viewed as the expected revenue at $(z, \theta)$.

**A5** *We assume that there exist $\kappa_1, \kappa_2 > 0$ such that for each $z \in \mathcal{Z}$, $\theta \in \Theta$, and $x \in \mathcal{X}$ we have*

$$\kappa_1 \max\left\{ (z^* - z)^2, \max_{1 \leq l \leq d} (\theta_0^{(\ell)} - \theta^{(l)})^2 \right\} \leq r(z^*, x, \theta_0) - r(z, x, \theta) \leq \frac{\kappa_2}{d+1} \|(z^* - z, \theta_0 - \theta)\|^2$$

*where $\|(z, \theta)\|^2 = \left( z^2 + \sum_{l=1}^d (\theta^{(l)})^2 \right)$.*

We conjecture that under assumptions A4, A2, and A5, a suitable modification to policy DEEP-C with Rounds would achieve a regret scaling similar to (if not the same as) that in Theorem 1. This conjecture rests on the following two key observations: (1) The optimal policy for the Oracle with adversarial covariates is the same as that under the i.i.d. covariates setting; and (2) policy DEEP-C with Rounds for i.i.d. covariates does not learn or use the distribution of $X_t$ (except via the knowledge of the constants $\alpha_2, \kappa_1$ and $\kappa_2$).

## E.2  Relaxing compactness of support of covariates

We believe that the compactness assumption of $\mathcal{X}$ in A1 can also be significantly relaxed. For example, consider the following simple relaxation. (We say that a random variable $W$ is $\sigma$-subgaussian if $\mathbb{P}(X > t) \leq e^{-\sigma^2 t^2}$.)

**A6** *$\{X_t\}_t$ and $\{Z_t\}_t$ are i.i.d. and mutually independent. Their distributions are unknown to the platform. The support of $Z_1$, namely $\mathcal{Z}$, is compact and known. Let*

$$W = \sup_{z \in \mathcal{Z}, \theta \in \Theta} z e^{\theta^\mathsf{T} X_1}.$$

*$W$ is $\sigma$-subgaussian for a known $\sigma > 0$.*

Under A2, and A3, and A6 we can obtain a result analogous to Theorem 1 as follows.

Recall that the policy DEEP-C with Rounds requires knowledge of $\alpha_2$, which in this case may be infinity. However, the platform can compute $\alpha_2'$ such that $P(W > \alpha_2') \leq 1/n^2$, and execute policy DEEP-C with $\alpha_2'$ instead of $\alpha_2$. Thus, the probability of event $\{\exists t \in \{1, \ldots, n\} V_t > \alpha_2'\}$ is at most $1/n$, and the overall impact of such an event on expected regret is $O(1)$. Using the fact that, since $\mathcal{Z}$ and $\Theta$ are compact, there exists $\alpha_1' > 0$ (possibly unknown to the platform) such that $P(W < \alpha_2') \leq 1/n^2$, we can obtain a regret bound similar to Theorem 1.