[Reviews · NeurIPS 2019]

Reviewer 1



*** The author rebuttal has sufficiently clarified my questions about the model assumptions, and the statements of A3 and theorem 1. *** Originality: The paper considers a model that combines censored feedback with contextual information and non-parametric residuals; the combination of all three has not been studied. They suggest a new algorithm and prove that it achieves logarithmic regret. Quality: Overall the results in the paper seem technically correct, but I had some concerns about the model/assumptions/results: - it is unclear to me how restrictive the parametric assumption that the expected log of the valuation is linear in the covariates. Although authors claim that predictor variables can be arbitrarily transformed, the task of designing the appropriate transformation, ie. "feature engineering", is a non-trivial task itself. Additionally it is unclear how realistic/flexible the logarithm/exponential relationship is, as it imposes that the valuations are exponentially far in relationship to the weighted covariate distances. If the purpose of the logarithmic form is to keep the valuation to be non-negative, there are other options for enforcing non-negativity as well. - Is there a typo in Assumption A3? the first inequality implies that if I were to take z = z^*, then r(z^*, \theta_0) = r(z^*, \theta) even for any \theta \neq \theta_0? Similarly the second inequality implies that if I were to take \theta = \theta_0, then r(z^*, \theta_0) = r(z, \theta_0) for any z \neq z^*? Is this a typo, or is this assumption as stated required and potentially restrictive? - In theorem 1, does the analysis assume a particular choice of the parameter \gamma, or does it hold for any nonnegative \gamma? As \gamma is used in the UCB/LCB, should we expect an expression similar to log(T)? I presume that if \gamma were chosen to be too small, then something in the regret bound would break so that you do not get logarithmic regret? Clarity: The paper reads clearly. Significance: The combination of censored feedback with contextual information and non-parametric residuals seems to be a practically relevant setup, although it is unclear how realistic/flexible the specific choice of the parametric model is. However the algorithm seems to be quite general and easily extendable to other parametric models as well. Misc question: - In your algorithm, you point out that each price selection checks one or more cells such that you get free exploration. In fact, you get additional information from knowing that if the user rejected the item at price p, then the user would have rejected the item at all prices higher than p. Therefore you get a datapoint even for cells that would have recommended higher prices than the chosen p_t. Symmetrically, if the user ended up buying the item at price p, then that means this user would have bought the item at all prices lower than p. Therefore we get a datapoint for cells that would have recommended lower prices than the chosen p_t. Do you imagine that the algorithm could be improved by further aggregating the data in this way? This may not be a good suggestion though as the inclusion of the datapoint to the estimate would be dependent on the the outcome of Y_t, which may cause biases.

Reviewer 2



I think the results presented represent solid progress on the problem and the results seem new, though there are a number of ways in which the results could be improved. * A major drawback of the DEEP-C algorithm is that it has exponential running time. Given that the authors work in a well-specified linear model, this is a bit surprising. Is there reason to believe that computational intractable should be expected for this setting? This would be good to discuss in the paper. If this is indeed the case, it would be nice to prove a computational lower bound. * For the regret bound in Theorem 1, the authors mention that regret scaling as $O(\sqrt{n})$ is necessary, so the bound is near-optimal in terms of dependence on $n$. However, the bound depends on a number of other parameters, and it would be good to discuss whether their dependence can be improved. Even better, it would be great to include a lower bound. EG, is regret scaling as $d^{11/4}$ to be expected? * Given that the authors are proposing a new model, they should spend a bit more time justifying the model and related assumptions and showing why it is natural, otherwise the whole paper feels like it not much more than a technical exercise. The fact that the model generalizes the non-contextual setup of Kleinberg and Leighton (2003) is a plus. Beyond this, the paper is fairly well-written and easy to follow. Some additional notes: * You may want to consider citing related work on semiparametric contextual bandits (Greenewald et al. 2017, Krishamurthy et al. 2018). * There are multiple places in the paper where the authors use the phrase "we conjecture..." in passing. If these are serious conjectures you should spend a bit more time to elaborate on these points and perhaps include an open problem section, otherwise I would recommend removing.

Reviewer 3



This is a good paper and I`d like to see it in the program. Both the model and the new algorithm are very valuable for the literature and they open interesting avenues of further enquiry (e.g. can we get log(T) bounds in this model if the noise distribution is benigh-enough). I have two concerns, though: 1. The algorithm runs in exponential time as there are n^{O(d)} buckets so the algorithm is not very practical unless the dimension is low. There are some techniques to obtain lower computational complexity using the algorithm of Plan and Vershynin, so this point is somewhat addressed in the paper, but this is a departure from the main technique of the paper. 2. The part that I am somewhat more concerned is that the paper seems to be re-deriving known results from stochastic bandits with side information. See for example “Leveraging side observations in stochastic bandits” (2012) or see the references in https://arxiv.org/pdf/1905.09898.pdf . I think none of those two are fatal problems and I still suggest the paper to be accepted despite this fact, but I strongly suggest the authors to address point 2 before publication. ---------- Post rebuittal: I am unconvinced that the literature on bandits with side information can’t be applied here. I understand that the authors prefer their customized approach, but the paper is very nice nevertheless and tying to a general framework is useful in the sense that it makes the paper more readable and helps others to build upon it. I strongly urge the authors to consider this more carefully. Here are my comments: (a) Your algorithm also discretizes the parameter space (even though you have a continuous ouput). There is no harm to also discretize the price up to 1/\sqrt{n} (so you at at most \sqrt{n} more regret) so you can imagine that each cell “suggests” a single price p(z,\theta,x) for each context X . So an arm is a policy parametrized by (z,\theta) that maps the context to a price. (b) the sequence graphs can be revealed in each period. Here the sequence graphs are given by the context. Two policies are linked in the graph for that period if they suggest the same price. (c) The action is also pulling an arm, selecting a policy and choosing the corresponding price. (d) The graphs links together policies that suggest the same price for the given context (note that the graph is different in each period, but you have a uniform bound on its independent set since the graph is a disjoint union of cliques, with each clique corresponding to a price so it is at most the number of prices)

[Author Response · NeurIPS 2019]

**Reviewer 1:**

*Model assumption:* We note that exponential sensitivity of the valuation with respect to covariate magnitudes can be resolved by using a logarithmic transformation of the covariates themselves. More generally, one may augment our approach with a machine learning algorithm which learns an appropriate transformation to fit the data well. Given these two observations the model is actually quite flexible, while admitting the learning guarantees we provide in the paper.

*Assumption A3:* Thanks for pointing that out. Indeed, that assumption is incorrectly restrictive – we apologize for the oversight on our part. Our results only need the following (weaker) assumption:

**A 3** *Let $\theta^{(l)}$ be the $l^{th}$ component of $\theta$, i.e., $\theta = (\theta^{(l)} : 1 \leq l \leq d)$. We assume that there exist $\kappa_1, \kappa_2 > 0$ such that for each $z \in \mathcal{Z}$ and $\theta \in \Theta$ we have*

$$\kappa_1 \max \left\{ (z^* - z)^2, \max_{1 \leq l \leq d} (\theta_0^{(\ell)} - \theta^{(l)})^2 \right\} \leq r(z^*, \theta_0) - r(z, \theta) \leq \frac{\kappa_2}{d+1} \|(z^* - z, \theta_0 - \theta)\|^2$$

*where $\|(z, \theta)\|^2 = \left( z^2 + \sum_{l=1}^{d} (\theta^{(l)})^2 \right)$.*

Our results and intermediate lemmas remain unchanged with the replacement of assumption A3 in the paper with the above revised assumption. By making minor appropriate changes in the Appendix, such as in line 473 and Assumption A5 before line 538, the technical correctness of the paper remains intact.

*Gamma:* Indeed, in the definition of DEEP-C with Rounds in the Appendix, in line 392 we assume that $\gamma = \max \left( 10\alpha_2^2, 4\frac{\kappa_2^2}{\log n}, \frac{\kappa_1^{-2}}{\log n} \right)$. We take your point and we will now mention this in the main paper. Note also that we discuss the role of $\gamma$ as a hyper-parameter to be tuned in the simulation sections.

*Side-information at higher prices:* Indeed using the feedback of rejection at a price for higher prices causes biases as you suggest, which seems challenging to deal with. Also, it is not clear if the gain of using such a feedback is significant enough in our setting. Note that the optimal algorithm for stochastic model in Klenberg-Leighton (2003), which is special case of our model with $d = 0$, also does not use such feedback.

**Reviewer 2:**

*Improving results for computational complexity and regret scaling w.r.t. d:* Regarding computational complexity, our paper suggests an algorithm (Sparse DEEP-C) that is more computationally efficient than DEEP-C, and our empirical study shows promising results for this approach. Certainly further work on the value of sparsity remains an important future direction.

*Lower bounds:* We note that even though we assume a linear parametric model, the residual distribution is assumed to be non-parametric. While Kleinberg and Leighton (2003) solved the problem for the non-contextual setting in the presence of censured (binary) feedback, which is a natural and important assumption for the dynamic pricing problem, simultaneously learning the residual distribution and $d$ dimensional parameters has remained an open problem, as evident from Table 1 in the paper. Our paper provides the first results at this level of generality. Also, as explained in our response to Reviewer 3, the structure of the problem does not meet the assumptions used in other general techniques such as bandits with side-information. While obtaining tight lower bounds on computational complexity and regret scaling w.r.t. number of dimensions $d$ for our setup would certainly be an interesting challenge for future work, we believe our current results already represent a significant progress for the dynamic pricing problem.

**Reviewer 3:** Thank you for your positive assessment.

*Computational complexity:* See response to reviewer 2 above.

*Relationship to bandits with side-information literature:* Thanks for pointing this out. Indeed there are some similarities between our work and this literature, in particular in our work too the reward information is revealed for a subset of arms where the subset may be a function of the chosen action. We will note this point in the final version of our paper.

However, to our knowledge, each paper on bandits with side information assumes (a) a discrete set of arms, (b) the existence of a sequence of graphs indexed by time (possibly fixed) with the arms as its nodes, (c) the action involves pulling an arm, and (d) at each time the reward at each neighbor of the pulled arm is revealed. Our model does not satisfy this structure. For example, if we view a cell $(i, j_1, \ldots, j_d)$ as an arm, then the cell corresponds to a subset of prices/actions. The subset of arms for which the reward is revealed at time $t$ depends on the covariate $x_t$, and the exact price $p_t$ from the above subset. Thus, the assumption of a pre-defined graph structure is not satisfied. Alternatively, if one views each $(z, \theta)$ as an arm then the set of arms is uncountable, and it thus does not meet the assumption of a discrete set of nodes. Thus, as far as we can tell, the results from this literature are not directly transferable to our setting. Also, note that none of the prior works in dynamic pricing literature (e.g., see the references in Table 1 in the paper) assume that the set of user-valuations/prices is finite, since a constant error in price leads to linear regret.

[Meta-Review · NeurIPS 2019]

The paper studies a variant of "contextual dynamic pricing", with a parametric model of buyer valuations and non-parametric "residual noise". The model is "rigged" in a specific way which allows for a strong positive result (and reviewers agree this is a feature, not a bug). Reviews are largely positive, but not without some caveats. As per one of the comments, relation to "bandits with side information" should be clarified.